

# Trends and source apportionment of aerosols in Europe during 1980–2018

Yang Yang[1], Sijia Lou[2*], Hailong Wang[3], Pinya Wang[1], Hong Liao[1]

[1]Jiangsu Key Laboratory of Atmospheric Environment Monitoring and Pollution Control, Jiangsu Collaborative Innovation Center of Atmospheric Environment and Equipment Technology, School of Environmental Science and Engineering, Nanjing University of Information Science and Technology, Nanjing, Jiangsu, China

[2]School of Atmospheric Sciences, Nanjing University, Nanjing, Jiangsu, China

[3]Atmospheric Sciences and Global Change Division, Pacific Northwest National Laboratory, Richland, Washington, USA

*Correspondence to slou.nju@gmail.com





**Abstract**
Aerosols have significantly affected health, environment and climate in Europe.
Aerosol concentrations have been declining since 1980s in Europe, mainly owing to
the reduction of local aerosol and precursor emissions. Emissions from other source
regions of the world, which have been changing rapidly as well, may also perturb the
historical and future trends of aerosols and change their radiative impact in Europe.
This study examines trends of aerosols in Europe during 1980–2018 and quantify
contributions from sixteen source regions using the Community Atmosphere Model
version 5 with an Explicit Aerosol Source Tagging technique (CAM5-EAST). The
simulated near-surface total mass concentration of sulfate, black carbon and primary
organic carbon had a 62% decrease during 1980–2018, of which the majority was
contributed by reductions of local emissions in Europe and 8%-9% was induced by
the decrease in emissions from Russia-Belarus-Ukraine. With the decreases in the
fractional contribution of local emissions, aerosols transported from other source
regions are increasingly important to air quality in Europe. During 1980–2018, the
decrease in sulfate loading leads to a warming effect of 2.0 W m$^{-2}$ in Europe, with
12% coming from changes in non-European sources, especially from North America
and Russia-Belarus-Ukraine. According to the Shared Socioeconomic Pathways
(SSP) scenarios, contributions to the sulfate radiative forcing over Europe from both
European local emissions and non-European emissions would decrease at a
comparable rate in the next three decades, suggesting that future changes in non-



European emissions are as important as European emissions in causing possible
regional climate change associated with aerosols in Europe.





## 1. Introduction


Aerosols are main air pollutants that contribute to excess morbidity and
premature mortality through damaging cardiovascular and respiratory systems
(Lelieveld et al., 2019). They also have adverse effects on atmospheric visibility for
road and air traffic (Vautard et al., 2009). During the 1952 London Fog, high fatality
associated with extremely high level of aerosols caused thousands of premature
deaths (Bell and Davis, 2001), which resulted in a number of air quality legislations
for reducing air pollution in Europe (Brimblecombe et al., 2006).
Besides the health and environment effects, aerosols can significantly impact
regional and global climate through perturbing the Earth's radiation fluxes and
influencing cloud microphysics (Boucher et al., 2013). Globally, anthropogenic
aerosols exert a net cooling effect in the Earth system, which have dampened the
warming driven by greenhouse gases since the pre-industrial era. Due to a strong
surface albedo feedback over polar regions, per unit aerosol emission from western
Europe was reported to have the greatest cooling effect than other major source
regions of the globe (Persad and Caldeira, 2018), revealing the importance of
understanding aerosol variations in Europe.
Significant reductions in near-surface aerosol concentrations and aerosol optical
depth (AOD) have been observed in Europe during the last few decades from long-
term station measurements and satellite retrievals (de Meij et al., 2012; Tørseth et
al., 2012). The decrease in aerosols has been considered as a cause of the increase
in surface solar radiation over Europe since the 1980s (Wild, 2009), as well as the



contributor of the eastern European warming (Vautard et al., 2009), Arctic
amplification (Acosta Navarro et al., 2016), and the increased atmospheric visibility
over Europe (Stjern et al., 2011) during the past three decades.
The decrease in aerosols over Europe was mainly attributed to the continuous
reductions in European local anthropogenic emissions of aerosols and precursor
gases since the 1980s (Smith et al., 2011), as a result of legislations for improving
air quality. In addition to local emissions, aerosol levels can also be affected by
aerosol transport at continental scales (Zhang et al., 2017; Yang et al., 2018a).
Aerosol emissions in major economic regions of the world have been changing
rapidly during the past few decades owing to economic development and
environmental measures. North America has started reducing emissions since the
1980s, and emissions in Russia also showed decreasing trends after the dissolution
of the Soviet Union (Smith et al., 2011). In the meantime, aerosol emissions from
East Asia and South Asia have largely increased due to economic growth, although
emissions in China have been undergoing a remarkable reduction in the most recent
years, as a result of strict air quality regulations (Streets et al., 2000; Li et al., 2017).
It is important to understand the relative roles of local emissions and regional
transport in affecting long-term variation of aerosols in Europe from both air quality
and climate perspectives.
Source apportionment is useful for quantifying contributions to aerosols from
specific source regions and/or sectors, which is beneficial to the emission control
strategies. The traditional method of examining the source-receptor relationship in



aerosol models is to zero out or perturb a certain percent of emissions from a given
source region or sector in parallel sensitivity simulations along with a baseline
simulation, which has been used in many studies to examine source contributions of
particulate matter (PM) in Europe from different sectors (e.g., Sartelet et al., 2012;
Tagaris et al., 2015; Aksoyoglu et al., 2016). Recently, source region contributions to
European CO and $O_3$ levels, as well as global and regional aerosol radiative forcing,
were examined under the Hemispheric Transport of Air Pollution model experiment
phase 2 (HTAP2) protocol, in which sensitivity simulations were conducted with
anthropogenic emissions from different source regions reduced by 20% (Jonson et
al., 2018). This method suffers a large computational cost for the excessive model
simulations when estimating contributions from a large number of sources, and
contributions from all sources do not sum up to 100% of the total concentration in the
default simulation (Koo et al., 2009; Wang et al., 2014).
The explicit aerosol tagging method, which simultaneously tracks contributions
from many different sources, is a useful tool for assessing source-receptor
relationship of aerosols. This method has previous been adopted in regional air
quality models such as CAMx (the Comprehensive Air quality Model with
Extensions) and CMAQ (the Community Multi-scale Air Quality model). Using
regional air quality models with aerosol tagging, contributions from different source
sectors and local/regional sources to European PM and its health impact were
studied (Brandt et al., 2013; Skyllakou et al., 2014; Karamchandani et al., 2017).
However, due to the limitation in domain size of regional air quality models,





contributions of intercontinental transport from sources outside the domain are
difficult to be accounted.
Anthropogenic emissions of aerosols and their precursor gases from different
economic regions of the world have changed substantially during the past few
decades. Very few studies have examined the source apportionment of aerosols in
Europe from sources all over the changing world. In this study, source attrubutions of
concentrations, column burden, optical depth of aerosols in four major areas of
Europe from sixteen source regions of the globe over 1980–2018 are quantified,
which is facilitated by the explicit aerosol source tagging technique that were recently
implemented in a global aerosol-climate model (CAM5-EAST). This technique has
lately been used to examine source attribution of aerosol trends in China and U.S.
during 1980–2014 (Yang et al., 2018a,b). The source apportionment analysis is
extended to year 2018 using the Shared Socioeconomic Pathways (SSPs) scenario,
with a focus on Europe here.
The CAM5-EAST model, along with the aerosol source tagging technique, and
aerosol emissions are described in Sect. 2. Section 3 evaluates the model
performance in simulating aerosols in Europe. Section 4 show the analysis of
source-receptor relationships of aerosols in Europe in climatological mean. Source
contributions to long-term variations of European aerosols and their direct radiative
forcing (DRF) during 1980–2018, as well as future forcing prediction, are provided in
Sect. 5. Section 6 summarizes these results and conclusions.
**2. Methods**


### 2.1 Model Description and Experimental Setup


The global aerosol-climate model CAM5 (Community Atmosphere Model version
5), which was developed as the atmospheric component of CESM (the Community
Earth System Model, Hurrell et al., 2013), is applied to simulate aerosols at a spatial
resolution of 1.9° latitude × 2.5° longitude and 30 vertical layers from the surface to
3.6 hPa. Aerosol species, including sulfate, black carbon (BC), primary organic
aerosol (POA), second organic aerosol (SOA), mineral dust and sea salt, can be
simulated in a modal aerosol module of CAM5. The three-mode aerosol module
(MAM3) configuration is chosen with the consideration of the computational
efficiency of long-term simulation. Details of the MAM3 aerosol representation in
CAM5 are described in Liu et al. (2012). On top of the default CAM5, some aerosol-
related scheme modifications are utilized to improve the model performance in the
aerosol convective transport and wet deposition (Wang et al., 2013).
A 40-year (1979–2018) historical AMIP-type (Atmospheric Model
Intercomparison Project) simulation has been performed, following CMIP6 (the
Coupled Model Intercomparison Project Phase 6) configurations and forcing
conditions. Time-varying sea surface temperatures, sea ice concentrations, solar
insolation, greenhouse gas concentrations and aerosol emissions are prescribed in
the simulation. To better reproduce large-scale circulation patterns for aerosol
transport in the model, wind fields are nudged to the MERRA-2 (Modern Era
Retrospective-Analysis for Research and Applications Version 2) reanalysis (Ronald
Gelaro et al., 2017).



Aerosol DRF is defined in this study as the difference in clear-sky radiative fluxes
at the top of the atmosphere between two parallel calculations in the radiative
transfer scheme with and without specific aerosols accounted, respectively.
Historical variation of aerosol DRF due to anthropogenic emissions from Europe and
outside Europe are quantified in this study. Future DRF of sulfate aerosol over
Europe is also estimated through scaling historical mean (1980–2018) sulfate DRF
by the ratio of SSPs future $SO_2$ emissions to historical emissions assuming a linear
response of DRF to AOD and regional emissions. This DRF prediction method has
been used to estimate the East Asian contribution to sulfate DRF in U.S. in 2030s
(Yang et al., 2018a).

**2.2 Aerosol Source Tagging Technique**

The Explicit Aerosol Source Tagging (EAST) technique, which was recently
implemented in CAM5 (Wang et al., 2014; Yang et al., 2017a, b), is used to examine
the long-term source apportionment of aerosols in Europe. Unlike the traditional
back-trajectory and emission perturbation methods, EAST has the identical physical,
chemical and dynamical processes considered independently for aerosol species
(defined as new tracers) emitted from each of the tagged source region and/or sector
in the simulation. Sulfate, BC, POA and SOA from pre-defined sources can be
explicitly tracked, from emission to deposition, in one CAM5-EAST simulation. Due
to the computational constraint and potentially large model bias from the simplified
SOA treatment (Yang et al., 2018a; Lou et al., 2019), we focus on sulfate, BC and
POA in this study but quantify the potential impact of SOA on the aerosol variation.


The global aerosol and precursor emissions are decomposed into sixteen source
regions defined in the HTAP2 protocol, including Europe (EUR), North America
(NAM), Central America (CAM), South America (SAM), North Africa (NAF), South
Africa (SAF), the Middle East (MDE), Southeast Asia (SEA), Central Asia (CAS),
South Asia (SAS), East Asia (EAS), Russia-Belarus-Ukraine (RBU), Pacific-
Australia-New Zealand (PAN), the Arctic (ARC), Antarctic (ANT), and Non-
Arctic/Antarctic Ocean (OCN) (Figure 1). Note that sources from marine and volcanic
eruptions are included in OCN. The focused receptor region in this study is Europe,
which is further divided into Northwestern Europe (NWE or NW Europe),
Southwestern Europe (SWE or SW Europe), Eastern Europe (EAE or E. Europe)
and Greece-Turkey-Cyprus (GTC) based on the finer source region selection in
HTAP2.
**2.3 Aerosol and Precursor Emissions**
Following the CMIP6-AMIP protocol, historical anthropogenic (Hoesly et al.,
2018) and biomass burning (van Marle et al., 2017) emissions of aerosol and
precursor gases are used over 1979–2014. For the remaining four years (2015–
2018), emissions are interpolated from the SSP2-4.5 forcing scenario, in which
aerosol pathways are not as extreme as other SSPs and have been used in many
model intercomparison projects for CMIP6 (O'Neill et al., 2016). Figure 2 shows the
spatial distribution and time series of anthropogenic emissions of $SO_2$ (precursor gas
of sulfate aerosol), BC and POA from Europe over 1980–2018. High emissions are
located over E. Europe and NW Europe, from which the emissions of $SO_2$, BC and





POA were reduced by 84–93%, 43–62% and 28–36%, respectively, in 2014–2018
relative to 1980–1984. Although SW Europe had a relatively low total amount of
emissions compared to E. Europe and NW Europe, it had significant reductions in
$SO_2$ and BC emissions, 91% and 55%, respectively. Over GTC region, $SO_2$ and BC
emissions were increased by 49% and 48%, respectively. Considering the sub-
regions as a whole, $SO_2$, BC and POA emissions from Europe have decreased by
12.57 Tg $yr^{-1}$ (83%), 0.22 Tg $yr^{-1}$ (46%) and 0.30 Tg $yr^{-1}$ (24%) in 2014–2018
compared to 1980–1984 (Table 1). Historical changes in emissions from other
source regions can be found in Hoesly et al. (2018) and Yang et al. (2018b).
**3 Model Evaluation**

Compared to the observational data from EMEP (European Monitoring and

Evaluation Programme, http://www.emep.int) networks during 2010–2014, CAM5-
EAST can well reproduce the spatial distribution and magnitude of aerosol
components with normalized mean biases (NMB) of -14%~-23% and correlation
coefficients (R) in a range of 0.43~0.62 for sulfate, BC and organic carbon (OC,
derived from POA and SOA from the model results) (Figures 3a, b, c). The model
underestimates the mean concentration of $PM_{2.5}$ (sum of sulfate, BC, POA and SOA)
by 59% relative to EMEP data (Figure 3d), although the spatial distribution has a
strong correlation with the observations (R=0.72). It is partially because the model
version used in this study does not have the ability to simulate nitrate and
ammonium aerosols, which can be the major constituents of $PM_{2.5}$ in some regions,





and the fine-mode mineral dust and sea salt is not included in the estimated $PM_{2.5}$
either.

Figure 4 shows the time series of annual mean near-surface sulfate, BC, OC and

$PM_{2.5}$ concentrations averaged over EMEP sites in Europe and the corresponding
model values during 1993–2018. Variations in near-surface sulfate and $PM_{2.5}$
concentrations are consistent between the model and observations, with R values
higher than 0.9. $PM_{2.5}$ concentrations are lower in the model simulation than
observations in almost all years, confirming the role of the missing aerosol species in
contributing to $PM_{2.5}$ as discussed above. The observed variations of BC and OC
concentrations in Europe are represented in the simulation, with R values of 0.52
and 0.65, respectively. However, the observed high values of BC and OC
concentrations are not captured by the model, probably because very few data were
available before 2010 and, therefore, any difference between model and observation
cannot be smoothed out through the spatial average. This is also indicated by the
large spatial variation of BC and OC concentrations before 2010. Nevertheless, the
modeled concentrations are still within the range of observations. Note that the
number of sites used for the spatial average in Figure 4 is different from year to year
and thus the variation or trend here does not represent that over a sub-region or the
entire Europe.

The modeled AOD is evaluated against the AERONET (Aerosol Robotic

Network, https://aeronet.gsfc.nasa.gov) data in Figure 8. Both the modeled and
observed AOD show decreasing trends during 2001–2018. The model



underestimates AOD in all four sub-regions of Europe probably due to the lack of
nitrate and ammonium aerosols. The variations of AOD in Western Europe
(combined NW and SW Europe) are well predicted with R values of about 0.75, but
the model barely reproduces the AOD variations in E. Europe and the GTC region
(R<0.5). The difference of the interannual variation in AOD between the model
simulation and observation can be caused by many factors such as aerosol
emissions, aerosol parameterizations in model, aerosol mixing state, inaccurate
meteorological fields from reanalysis data, and biases in measurements. However,
identifying the contribution of each factor to the difference is beyond the scope of this
paper.
**4. Source Apportionment of Aerosols in Europe**
Based on the tagging technique in CAM5-EAST, near-surface concentrations of
total sulfate-BC-POA can be attributed to emissions within and outside Europe, as
shown in Figures 5a and 5b, and the relative contributions in percentage are given in
Figures 5c and 5d. Averaged over 2010–2018, due to the relatively high local
emissions, annual mean sulfate-BC-POA concentrations contributed by European
emissions show peak values of 4 μg m$^{-3}$ in E. Europe. The slight increase in $SO_2$
emission from the GTC region (Figure 2), which is opposite to the decreases in the
other three sub-regions of Europe, also leads to high concentrations in GTC, with 2–
4 μg m$^{-3}$ contributed by European emissions. Due to the atmospheric transport from
surrounding regions including North Africa, the Middle East and RBU, non-European
emissions account for 0.5–1 μg m$^{-3}$ over SW Europe, E. Europe and GTC area.



Overall, European local emissions are the dominant sources of sulfate-BC-POA
near-surface concentrations in Europe with contributions larger than 80% (60%) in
central areas (most of Europe). Non-European emissions are responsible for 30–
50% of the near-surface concentrations near the coastal areas and boundaries of the
Europe that are easily influenced by aerosol regional transport.
Figure 6 illustrates the source contributions in percentage of emissions from
different regions of the globe to near-surface aerosol concentrations and column
burdens over the four sub-regions of Europe averaged over 2010–2018. Different
aerosols have fairly different local/remote source attributions depending on the local
to remote emission ratio and transport efficiency. European emissions explain 54%–
68% of near-surface sulfate concentrations over the four sub-regions of Europe, with
the largest local contribution from E. Europe due to the relatively high emission rate.
The emissions from Europe dominate BC and POA concentrations in Europe with
contributions in the range of 78%–95% and 58%–78%, respectively. The impact of
local emissions on near-surface sulfate concentration is smaller than BC and POA.
This is partially due to the less efficient gas scavenging than particles and the
additional $SO_2$-to-sulfate conversion process that increases the atmospheric
residence time of sulfur. On the other hand, the higher initial injection height of $SO_2$
emissions from some sources (e.g., industrial sector and power plants) facilitates the
lifting of $SO_2$ and sulfate aerosol into the free atmosphere and, therefore, favors the
long-range transport (Yang et al., 2019). The efficient reduction in local $SO_2$



emissions in Europe also caused the lower influences of local emissions on sulfate
concentrations in recent years.

Anthropogenic emissions over oceans (e.g., international shipping) and natural

emissions of oceanic dimethyl sulfide (DMS) and volcanic activities together account
for 16%–28% of near-surface sulfate concentrations in the four sub-regions of
Europe. About 10% of sulfate and 5%–10% of BC and POA in E. Europe and GTC
come from RBU emissions. North Africa contributes to 7% of sulfate, 17% of BC and
24% of POA over SW Europe. The contributions of emissions, from the Middle East,
to aerosol concentrations in GTC are between 5% and 10%.

The transboundary and intercontinental transport of aerosols occur most

frequently in the free troposphere rather than near the surface, leading to larger
relative contributions from non-European sources to aerosol column burdens than to
the near-surface concentrations (Figure 6). The European emissions only contribute
32%–47% of column burden of sulfate, 57%–75% of BC and 51%–71% of POA over
the four sub-regions of Europe. Over NW Europe and SW Europe, about 10%–15%
of the sulfate burden is from East Asia and RBU, respectively. Sources in North
Africa are responsible for 27% and 14% of BC and 19% and 11% of POA burden
over SW Europe and GTC, respectively. Emissions from North America account for
15% and 10% POA burden over NW Europe and SW Europe, respectively.
Emissions from RBU explain 12% and 19% of POA burden over E. Europe and
GTC, respectively. Since near-surface aerosol concentrations directly affect air
quality and column burden is more relevant to radiative impact, the differences of



relative contributions between near-surface concentrations and column burden
highlight the possible roles of non-local emissions in either air quality or energy
balance over Europe.
**5. Source Apportionment of Long-term Trends**
Total sulfate-BC-POA concentrations decreased during 1980–2018 over all of
the four sub-regions of Europe (Figure 7), since that near-surface aerosol
concentrations in Europe are dominated by its local emissions and the European
anthropogenic emissions have significantly decreased during this time period.
Averaged over the entire Europe, near-surface concentrations of sulfate, BC and
POA decreased by 70%, 43% and 23%, respectively, between 1980–1984 and
2014–2018, which is consistent with the decreases in local emissions (Table 1). The
total sulfate-BC-POA concentrations decreased by 62%. With SOA included, this
value does not have a substantial change (from 62% to 59%) and the decreasing
trends in the four sub-regions of the Europe are not largely affected either. The
column burden of sulfate, BC, POA and the sum of these three decreased by 60%,
28%, 4% and 55%, respectively, which is less than the decrease in corresponding
near-surface concentration. It is because non-local emissions have larger influences
at high altitudes than at the surface, which partly dampened the contribution of near-
surface aerosol decrease (induced by reductions in location emissions) to the
column integration.
The decrease in European local emissions explains 93% of the reduced
concentration and 91% of the reduced burden in Europe between the first and last


five-year period of 1980–2018, while 8%–9% is contributed by the reduction in
emissions from RBU (Table 2). The decrease in emissions from North America also
explains 10% of the reduced column burden of sulfate-BC-POA in Europe from
1980–1984 to 2014–2018. Along with the decreases in local emission contributions
to near-surface sulfate-BC-POA concentrations in Europe, the fraction of non-
European emission contributions increased from 10%–30% to 30%–50% during
1980–2018 (Figure 7), indicating that aerosols from foreign emissions through long-
range transport have become increasingly important to air quality in Europe.
Regulations for further improvement of air quality in Europe in the near future need
to take changes in non-European emissions into account.

Similar to the declining trend in column burden, simulated total AOD also

decreased from 0.12–0.16 to 0.06–0.08 in NW Europe and SW Europe and from
0.19–0.21 to 0.09–0.13 in E. Europe and GTC region during the past four decades
(Figure 8). Sulfate AOD accounts for the largest portion of total combustion AOD
(sum of sulfate, BC, POA and SOA) over the four sub-regions of Europe. The
combustion AOD has decreased by 0.065 from 1980–1984 to 2014–2018 (Table 1),
with 0.059 (91%) contributed by the decrease in sulfate AOD. Therefore, we focus
on sulfate aerosol when examining the decadal changes in AOD and DRF in Europe
below.

The decreased sulfate AOD can also be decomposed into different contributions

from individual source regions in CAM5-EAST. European local emissions contribute
to 89% of the decrease, followed by 9% and 7% attributed to changes in emissions





from RBU and North America, respectively, with the residual offset by other source
regions (Table 2). Over the last four decades, model simulated sulfate AOD
decreased at a rate of 0.017, 0.017, 0.026 and 0.012 decade$^{-1}$, respectively, over
NW Europe, SW Europe, E. Europe and GTC. Decreases in European local $SO_2$
emissions result in 78% of the sulfate AOD decreases over GTC and about 90%
over the other three sub-regions (Figure 9). For the remote sources, emission
changes in North America explain 5%–10% of the European sulfate AOD decrease,
while RBU sources contribute 29% of the sulfate AOD decrease over GTC and 6%–
8% over NW Europe and E. Europe, indicating a possible warming enhancement
effect of changes in emissions from North America and RBU.

Averaged over 1980–2018, sulfate imposed a cooling effect over Europe with the

maximum negative DRF at the top of the atmosphere (TOA) exceeding –3 W m$^{-2}$ in
E. Europe (Figure 10). Compared to 1980–1984, the magnitude of sulfate DRF
decreased in 2014–2018, leading to a 1–3 W m$^{-2}$ warming mainly in E. Europe. The
warming effect mostly came from local $SO_2$ emission reduction, while non-European
emission changes only contributed less than 0.4 W m$^{-2}$ over most regions of the
Europe. Considering Europe as a whole, the decrease in sulfate DRF caused a
warming effect of 2.0 W m$^{-2}$, with 88% and 12% coming from reductions European
local emissions and changes in non-European emissions, respectively (Tables 1 and

2).

Future changes in sulfate DRF associated with European and non-European

emissions based on eight SSP scenarios are also estimated and shown in Figure 11.



Sulfate DRF contributed by both European and non-European emissions would
decrease in the near future but has large variabilities between different SSPs. The
sulfate DRF (cooling) over Europe contributed from European local emissions shows
a decrease from -0.48 W m$^{-2}$ in year 2015 to -0.18 (-0.08 ~ -0.33) W m$^{-2}$ in year 2030
and -0.14 (-0.05 ~ -0.29) W m$^{-2}$ in year 2050. Unlike their contributions to the
historical (1980–2018) change, non-European emissions have an increasingly
significant impact on the future sulfate DRF changes in Europe. The contributions of
non-European emissions decrease from -0.68 W m$^{-2}$ in year 2015 to -0.39 (-0.13 ~ -
0.64) W m$^{-2}$ in year 2030 and -0.26 (-0.08 ~ -0.63) W m$^{-2}$ in year 2050, with the
changes in a magnitude similar to that of European local emissions. It suggests that
future changes in non-European emissions are as important as European emissions
to radiative balance and associated regional climate change in Europe.
**6. Conclusions**

Using a global aerosol-climate model with an explicit aerosol source tagging

technique (CAM5-EAST), we examine the long-term trends and source
apportionment of aerosols in Europe over 1980-2018 from sixteen source regions
covering the globe in this study. CAM5-EAST can well capture the spatial distribution
and temporal variation of aerosol species in Europe during this time period, although
it underestimates PM$_{2.5}$ concentration and total AOD due in part to the lack of
representation of nitrate and ammonium aerosols in the model.

Averaged over 2010–2018, European emissions account for 54%–68%, 78%–

95% and 58%–78% of near-surface sulfate, BC, and POA concentrations over



Europe, respectively. RBU emissions explain 10% of sulfate in E. Europe and GTC.
North Africa contributes to 17% of BC and 24% of POA over SW Europe.
Anthropogenic emissions over oceans (e.g., from international shipping) and natural
emissions from marine and volcanic activities together account for 16%–28% of
sulfate near-surface concentrations in Europe. European emissions only account for
32%–47%, 57%–75% and 51%–71% of column burden of sulfate, BC and POA,
respectively, in Europe, with the rest contributed by emissions from East Asia, RBU,
North Africa and North America.

Compared to 1980–1984, simulated total sulfate-BC-POA near-surface

concentration and column burden over 2014–2018 had a decrease of 62% and 55%,
respectively, the majority of which was contributed by reductions in European local
emissions. The decrease in emissions from RBU contributed 8%–9% of the near-
surface concentration decrease, while the decrease in emissions from North America
accounted for 10% of the reduced column burden. With the large decrease in local
emission contribution, aerosols from foreign sources became increasingly important
to air quality in Europe. The decrease in sulfate led to a 2.0 W m$^{-2}$ warming in
Europe, with 12% coming from changes in non-European emissions, especially in
North America and RBU. Based on the SSP scenarios and the assumed relationship
between DRF and emissions, we estimated that sulfate DRF over Europe
contributed from European emissions and non-European emissions would decrease
at a comparable rate in the near future. This suggests that future changes in non-
European emissions are as important as European emissions in affecting regional



climate change associated with aerosols in Europe. It should also be noted that the
model currently does not have the ability to simulate nitrate and ammonium aerosols
and, therefore, the conclusions may not hold with all aerosols.






***Data availability.***
The default CAM5 model is publicly available at
http://www.cesm.ucar.edu/models/cesm1.2/ (last access: 16 August 2019). Our
CAM5-EAST model code and results can be made available through the National
Energy Research Scientific Computing Center (NERSC) servers upon request.

***Competing interests.***
The authors declare that they have no conflict of interest.

***Author contribution***.
YY, SL, and HW designed the research; YY performed the model simulations; YY,
and SL analyzed the data. All the authors discussed the results and wrote the paper.

***Acknowledgments.***
This research was support by the National Natural Science Foundation of China
under grant 41975159, the U.S. Department of Energy (DOE), Office of Science,
Biological and Environmental Research as part of the Earth and Environmental
System Modeling program, Jiangsu Specially Appointed Professor Project, and the
Startup Fund for Talent at NUIST under Grant 2019r047. The Pacific Northwest
National Laboratory is operated for DOE by Battelle Memorial Institute under
contract DE-AC05-76RLO1830. The National Energy Research Scientific Computing
Center (NERSC) provided computational support.



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

Contributions of local and regional sources to fine PM in the megacity of Paris,
Atmos. Chem. Phys., 14, 2343– 2352, https://doi.org/10.5194/acp-14-2343-
2014, 2014.
Smith, S. J., van Aardenne, J., Klimont, Z., Andres, R. J., Volke, A., and Delgado
Arias, S.: Anthropogenic sulfur dioxide emissions: 1850–2005, Atmos. Chem.
Phys., 11, 1101-1116, https://doi.org/10.5194/acp-11-1101-2011, 2011.
Stjern, C. W., Samset, B. H., Myhre, G., Bian, H., Chin, M., Davila, Y., Dentener, F.,
Emmons, L., Flemming, J., Haslerud, A. S., Henze, D., Jonson, J. E., Kucsera,
574          T., Lund, M. T., Schulz, M., Sudo, K., Takemura, T., and Tilmes, S.: Global and
regional radiative forcing from 20 % reductions in BC, OC and SO4 – an HTAP2
multi-model study, Atmos. Chem. Phys., 16, 13579–13599,
https://doi.org/10.5194/acp-16-13579-2016, 2016.




Stjern, C. W., Stohl, A., and Kristjánsson, J. E.: Have aerosols affected trends in
visibility and precipitation in Europe?, J. Geophys. Res., 116, D02212,
https://doi.org/10.1029/2010JD014603, 2011.

Streets, D. G., Tsai, N. Y., Akimoto, H., and Oka, K.: Sulfur dioxide emissions in Asia
in the period 1985–1997, Atmos. Environ., 34, 4413–4424,
https://doi.org/10.1016/S1352-2310(00)00187-4, 2000.

Tagaris, E., Sotiropoulou, R., Gounaris, N., Andronopoulos, S., and Vlachogiannis,
D.: Effect of the Standard Nomenclature for Air Pollution (SNAP) categories on
air quality over Europe, Atmosphere, 6, 1119, doi:10.3390/atmos6081119, 2015.

Tørseth, K., Aas, W., Breivik, K., Fjæraa, A. M., Fiebig, M., Hjellbrekke, A. G., Lund
Myhre, C., Solberg, S., and Yttri, K. E.: Introduction to the European Monitoring
and Evaluation Programme (EMEP) and observed atmospheric composition
change during 1972–2009, Atmos. Chem. Phys., 12, 5447-5481,
https://doi.org/10.5194/acp-12-5447-2012, 2012.

van Marle, M. J. E., Kloster, S., Magi, B. I., Marlon, J. R., Daniau, A.-L., Field, R. D.,
Arneth, A., Forrest, M., Hantson, S., Kehrwald, N. M., Knorr, W., Lasslop, G., Li,
F., Mangeon, S., Yue, C., Kaiser, J. W., and van der Werf, G. R.: Historic global
biomass burning emissions for CMIP6 (BB4CMIP) based on merging satellite
observations with proxies and fire models (1750–2015), Geosci. Model Dev., 10,
3329–3357, https://doi.org/10.5194/gmd-10-3329-2017, 2017.

Vautard, R., Yiou, P., and Oldenborgh, G.: Decline of fog, mist and haze in Europe
over the past 30 years, Nat. Geosci., 2, 115–119,
https://doi.org/10.1038/ngeo414, 2009.

Wang, H., Easter, R. C., Rasch, P. J., Wang, M., Liu, X., Ghan, S. J., Qian, Y., Yoon,
J.-H., Ma, P.-L., and Vinoj, V.: Sensitivity of remote aerosol distributions to
representation of cloud–aerosol interactions in a global climate model, Geosci.
Model Dev., 6, 765–782, https://doi.org/10.5194/gmd-6-765-2013, 2013.

Wang, H., Rasch, P. J., Easter, R. C., Singh, B., Zhang, R., Ma, P.-L., Qian, Y.,
Ghan, S. J., and Beagley, N.: Using an explicit emission tagging method in
global modeling of source-receptor relationships for black carbon in the Arctic:
Variations, sources, and transport pathways, J. Geophys. Res.-Atmos., 119,
12888–12909, https://doi.org/10.1002/2014JD022297, 2014.

Wild, M.: Global dimming and brightening: A review, J. Geophys. Res., 114,
D00D16, https://doi.org/10.1029/2008JD011470, 2009.



Yang, Y., Wang, H., Smith, S. J., Ma, P.-L., and Rasch, P. J.: Source attribution of
black carbon and its direct radiative forcing in China, Atmos. Chem. Phys., 17,
4319-4336, https://doi.org/10.5194/acp-17-4319-2017, 2017a.
Yang, Y., Wang, H., Smith, S. J., Easter, R., Ma, P.-L., Qian, Y., Yu, H., Li, C., and
Rasch, P. J.: Global source attribution of sulfate concentration and direct and
indirect radiative forcing, Atmos. Chem. Phys., 17, 8903-8922,
https://doi.org/10.5194/acp-17-8903-2017, 2017b.
Yang, Y., Wang, H., Smith, S. J., Zhang, R., Lou, S., Yu, H., Li, C., and Rasch, P. J.:
Source apportionments of aerosols and their direct radiative forcing and long-
term trends over continental United States, Earth's Future, 6, 793–808,
https://doi.org/10.1029/2018EF000859, 2018a.
Yang, Y., Wang, H., Smith, S. J., Zhang, R., Lou, S., Qian, Y., Ma, P.-L., and Rasch,
P. J.: Recent intensification of winter haze in China linked to foreign emissions
and meteorology, Sci. Rep., 8, 2107, https://doi.org/10.1038/s41598-018-20437-
7, 2018b.
Yang, Y., Smith, S. J., Wang, H., Lou, S., and Rasch, P. J.: Impact of anthropogenic
emission injection height uncertainty on global sulfur dioxide and aerosol
distribution, J. Geophys. Res.-Atmos., 124, 4812–4826.
https://doi.org/10.1029/2018JD030001, 2019.
Zhang, Q., Jiang, X., Tong, D., Davis, S. J., Zhao, H., Geng, G., Feng, T., Zheng, B.,
Lu, Z., Streets, D. G., Ni, R., Brauer, M., van Donkelaar, A., Martin, R. V., Huo,
H., Liu, Z., Pan, D., Kan, H., Yan, Y., Lin, J., He, K., and Guan, D.:
Transboundary health impacts of transported global air pollution and
international trade, Nature, 543, 705–709, https://doi.org/10.1038/nature21712,
2017.


**Table 1.** Annual emissions (Tg yr$^{-1}$), concentrations (μg m$^{-3}$), column burden (mg m$^{-2}$), AOD (scaled up by a factor of 100) and DRF (W m$^{-2}$) of Sulfate, BC, POA, SBP (sulfate-BC-POA) and PM$_{2.5}$ (sulfate-BC-POA-SOA) in Europe averaged over 1980–1984 and 2014–2018, as well as the differences between 1980–1984 and 2014–2018. Differences in percentage relative to mean values in 1980–1984 are presented in parentheses.

|  |  | Emis. | Conc. | Burden | AOD*100 | DRF |
|---|---|---|---|---|---|---|
| Sulfate | 1980–1984 | 15.10 | 6.00 | 14.35 | 9.13 | -3.27 |
|  | 2014–2018 | 2.53 | 1.80 | 5.79 | 3.24 | -1.24 |
|  | Δ | -12.57 (-83.2) | -4.20 (-70.0) | -8.55 (-59.6) | -5.89 (-64.6) | 2.04 (-62.2) |
| BC | 1980–1984 | 0.47 | 0.4 | 0.38 | 0.7 | -- |
|  | 2014–2018 | 0.25 | 0.23 | 0.28 | 0.5 | -- |
|  | Δ | -0.22 (-45.8) | -0.17 (-43.0) | -0.11 (-27.6) | -0.21 (-29.2) | -- |
| POA | 1980–1984 | 1.24 | 1.12 | 1.12 | 0.63 | -- |
|  | 2014–2018 | 0.94 | 0.86 | 1.08 | 0.58 | -- |
|  | Δ | -0.30 (-24.4) | -0.26 (-23.2) | -0.04 (-3.8) | -0.05 (-7.5) | -- |
| Sulfate-BC-POA | 1980–1984 | -- | 7.52 | 15.85 | 10.46 | -- |
|  | 2014–2018 | -- | 2.89 | 7.15 | 4.32 | -- |
|  | Δ | -- | -4.63 (-61.6) | -8.70 (-54.9) | -6.15 (-58.7) | -- |
| PM$_{2.5}$ | 1980–1984 | -- | 10.48 | 19.58 | 11.92 | -- |
|  | 2014–2018 | -- | 4.34 | 8.55 | 5.44 | -- |
|  | Δ | -- | -6.14 (-58.6) | -11.03 (-56.3) | -6.48 (-54.37) | -- |




**Table 2.** Relative contributions (%) of emissions from major source regions to the
changes in near-surface concentrations, column burden, AOD and DRF in Europe
between 1980–1984 and 2014–2018.

| | Sulfate-BC-POA | | | |
|---|---|---|---|---|
| | Δ Conc. | Δ Burden | Δ AOD | |
| EUR | 92.8 | 91.2 | 91.2 | |
| NAM | 1.8 | 10.0 | 6.5 | |
| NAF | -1.0 | -1.5 | -1.6 | |
| MDE | -0.9 | -1.9 | -1.5 | |
| EAS | -0.3 | -3.1 | -1.7 | |
| RBU | 8.0 | 9.2 | 8.5 | |
| OTH | -0.1 | -4.2 | -2.0 | |
| OCN | -0.3 | 0.2 | 0.6 | |
| | Sulfate | | | |
| | Δ Conc. | Δ Burden | Δ AOD | Δ DRF |
| EUR | 91.3 | 89.2 | 88.9 | 88.2 |
| NAM | 2.1 | 10.5 | 6.9 | |
| NAF | -0.6 | -0.9 | -0.8 | |
| MDE | -0.8 | -1.7 | -1.3 | |
| EAS | -0.3 | -2.8 | -1.4 | 11.8 |
| RBU | 8.6 | 9.5 | 8.7 | |
| OTH | -0.1 | -4.0 | -1.8 | |
| OCN | -0.3 | 0.3 | 0.7 | |




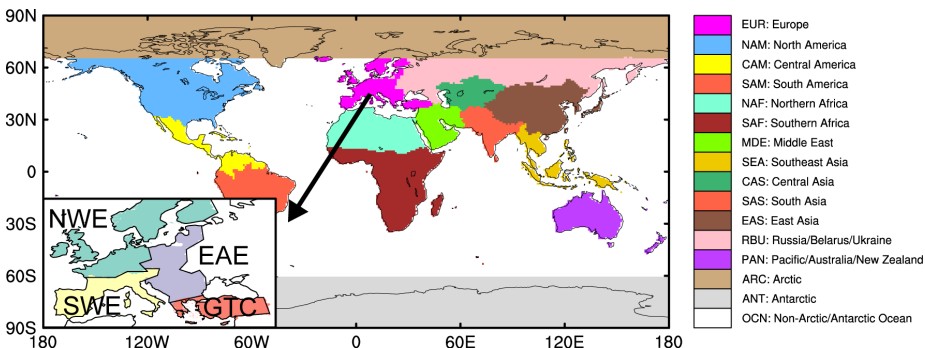



**Figure 1.** Source regions that are selected for the Explicit Aerosol Source Tagging
(EAST) in this study, including Europe (EUR), North America (NAM), Central
America (CAM), South America (SAM), North Africa (NAF), South Africa (SAF), the
Middle East (MDE), Southeast Asia (SEA), Central Asia (CAS), South Asia (SAS),
East Asia (EAS), Russia-Belarus-Ukraine (RBU), Pacific-Australia-New Zealand
(PAN), the Arctic (ARC), Antarctic (ANT), and Non-Arctic/Antarctic Ocean (OCN).
The embedded panel (at bottom left) is Europe, as the receptor region, which is
further divided to Northwestern Europe (NWE), Southwestern Europe (SWE),
Eastern Europe (EAE) and Greece-Turkey-Cyprus (GTC).



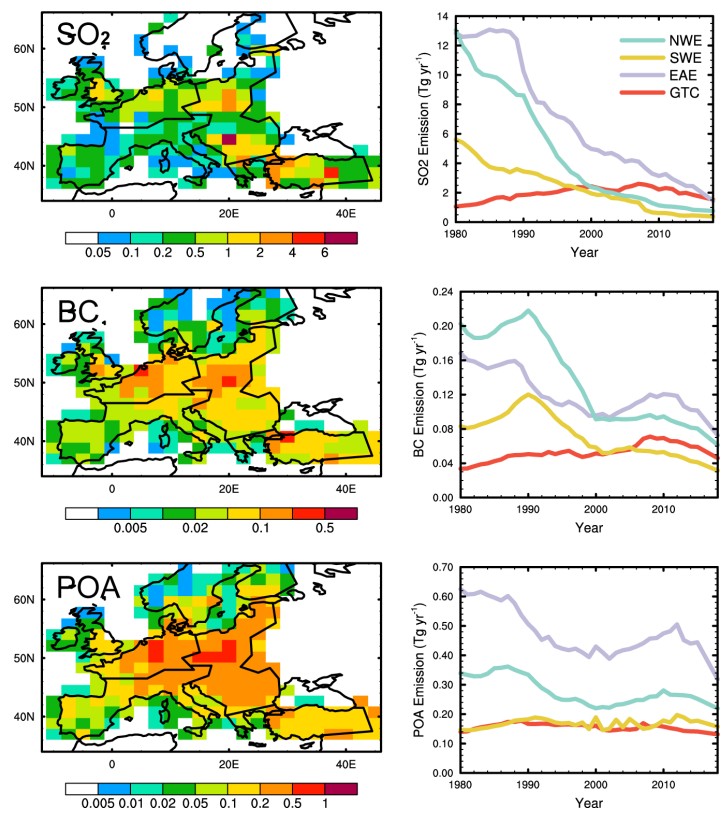

**Figure 2.** Spatial distribution (left) of annual mean (1980–2018) SO₂ (top), BC (middle) and POA (bottom) emissions (Tg m$^{-2}$ yr$^{-1}$) over Europe. Time series (1980–2018) of annual total SO₂, BC and POA emissions (Tg yr$^{-1}$) from the four sub-regions of Europe.



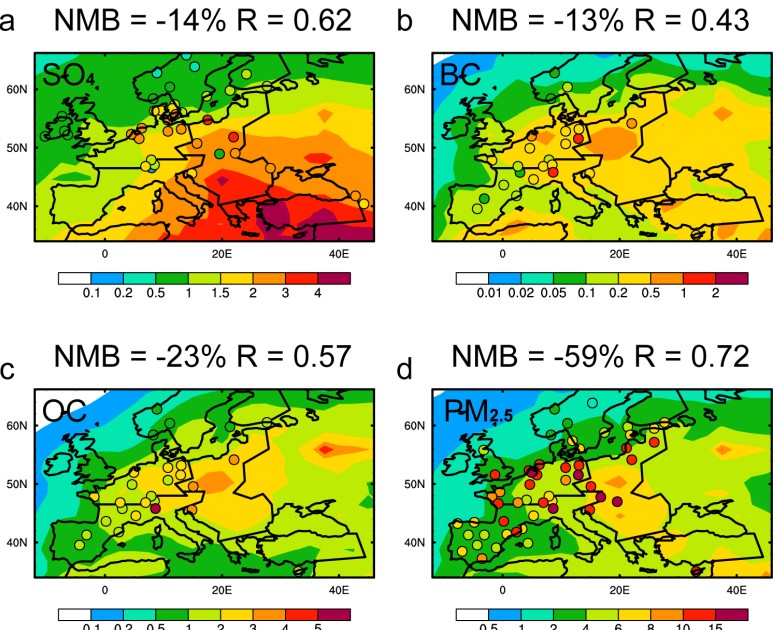

**Figure 3.** Spatial distribution of simulated (contour) and observed (color-filled circles) annual mean near-surface (a) sulfate, (b) BC, (c) OC (derived as (POA+SOA)/1.4 in model) and (d) $PM_{2.5}$ (sulfate+BC+POA+SOA in model) concentrations (μg m$^{-3}$) over 2010–2014. Observations are from EMEP (European Monitoring and Evaluation Programme) networks. Normalized mean bias ( NMB = $100\% \times \sum (Model_{site} - Observation_{site}) / \sum Observation_{site}$ ) and correlation coefficient (R) between observed and simulated concentrations are noted at the top of each panel.



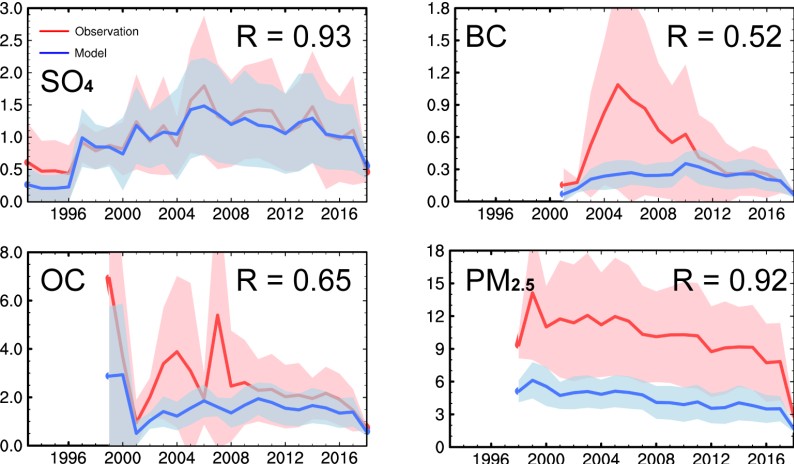

**Figure 4.** Time series (1993–2018) of spatial and annual mean near-surface sulfate, BC, OC and PM$_{2.5}$ concentrations (µg m$^{-3}$) in Europe from model simulation (blue lines) and observations (red lines). Model results are plotted only when EMEP observational data are available. Shaded areas represent 1-σ spatial standard deviation of annual mean concentrations for each year. Temporal correlation coefficients (R) between observed and simulated spatially averaged concentrations are noted on the top-right corner of each panel.

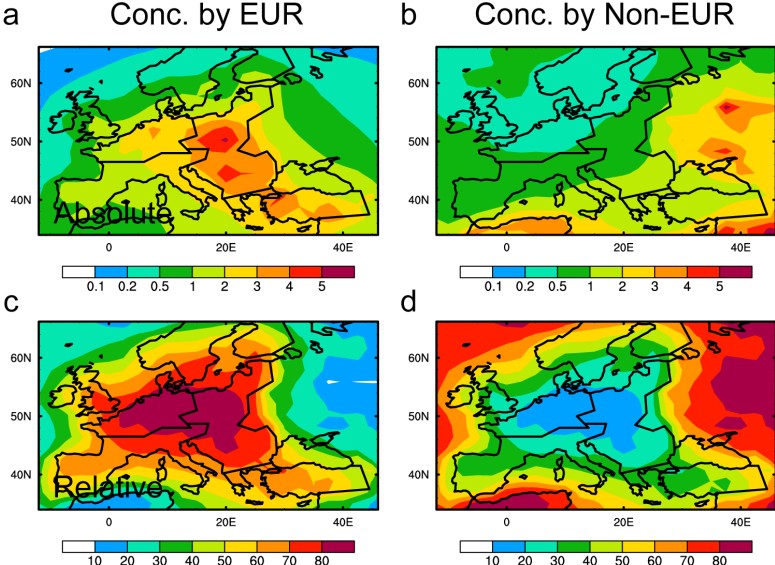

**Figure 5.** (a,b) Absolute (µg m$^{-3}$) and (c,d) relative contributions (%) to annual mean near-surface concentrations of sulfate-BC-POA from European local emissions and emissions outside the Europe, respectively, averaged over 2010–2018.





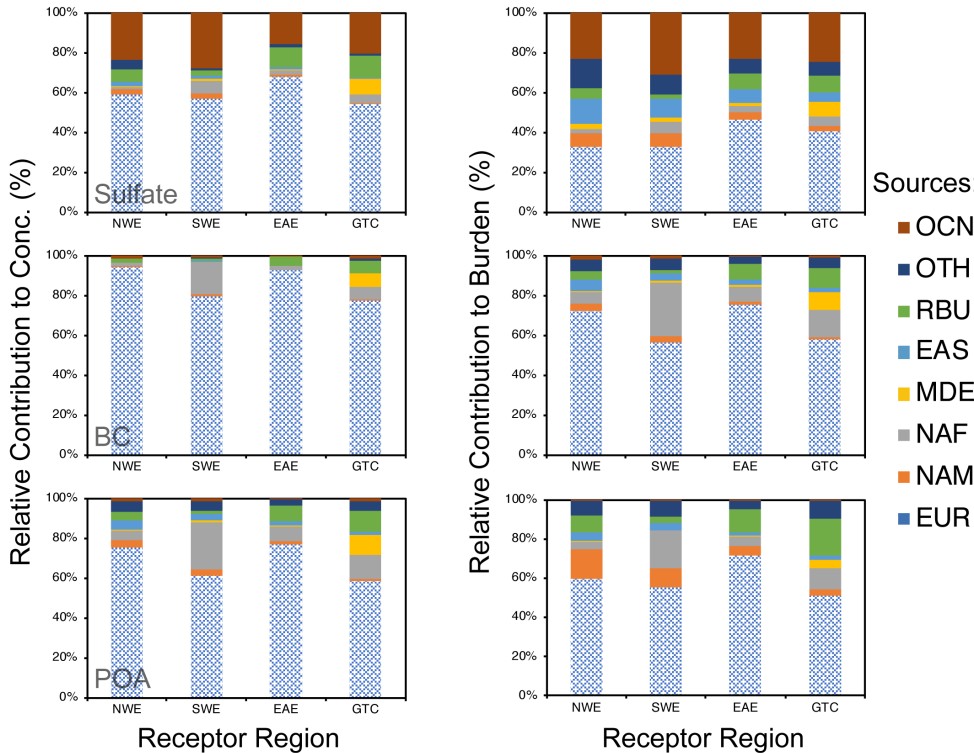

**Figure 6.** Relative contributions (%) by emissions from major tagged source regions
(EUR, NAM, NAF,MDE, EAS, RBU, OCN) and other regions
(OTH=CAM+SAM+SAF+SEA+CAS+SAS+PAN+ARC+ANT) to near-surface
concentrations (left) and column burdens (right) of sulfate, BC and POA (from top to
bottom) in the four sub-regions of Europe averaged over 2010–2018. Patterned
areas represent EUR local contributions.

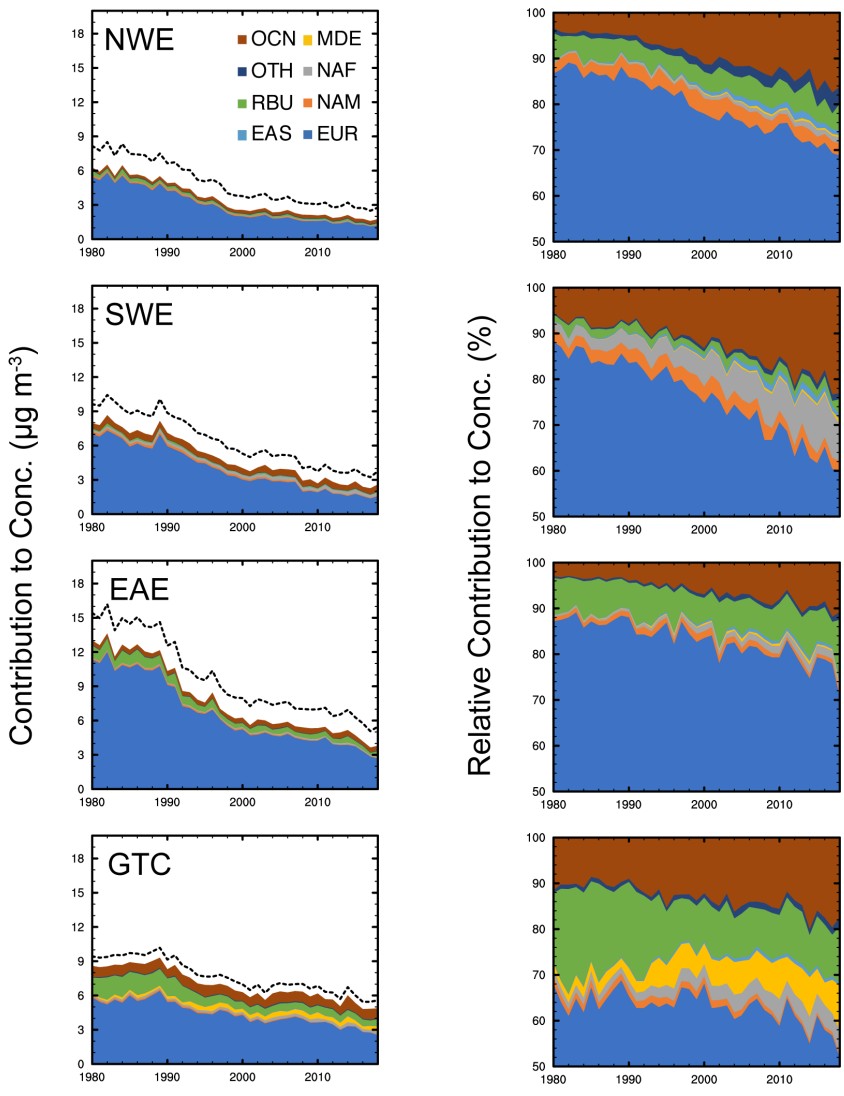

**Figure 7.** Time series (1980–2018) of absolute (left, µg m⁻³) and relative (right, %) contributions of emissions from major source regions to the simulated annual mean near-surface sulfate-BC-POA concentrations averaged over the four sub-regions of Europe. Dashed lines in left panels represent simulated aerosol concentrations including SOA.





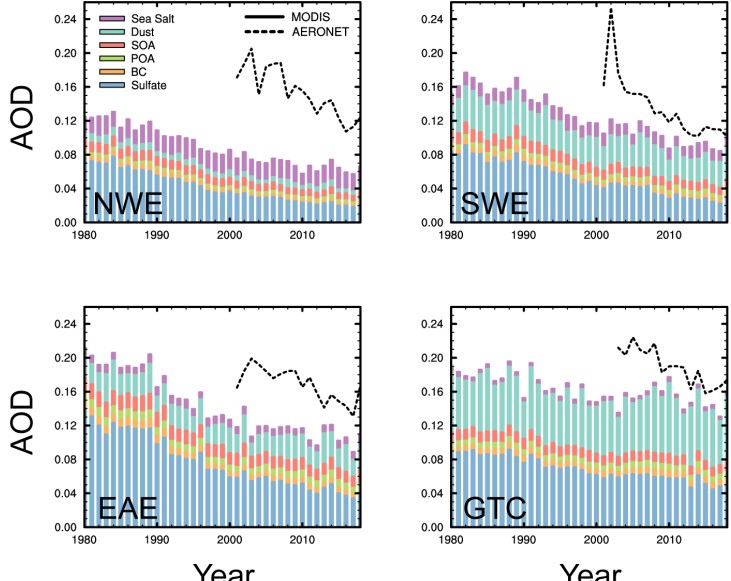



**Figure 8.** Time series (1980–2018) of simulated annual mean AOD for sulfate, BC,
POA, SOA, dust and sea salt averaged over the four sub-regions of Europe. Dashed
lines represent AOD from AERONET measurements.






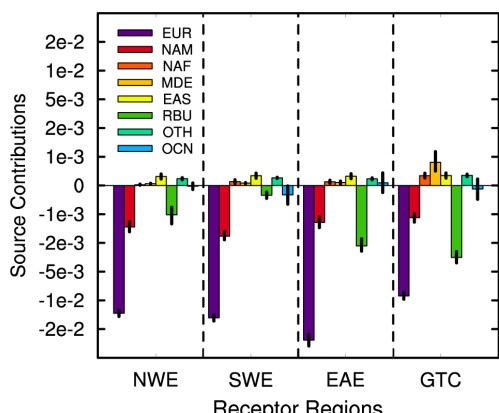

**Figure 9.** Absolute contributions (decade[-1]) of the emissions from major source regions to the trends of sulfate AOD over the four sub-regions of Europe. Error bars represent 95% confidence intervals of the linear regression.

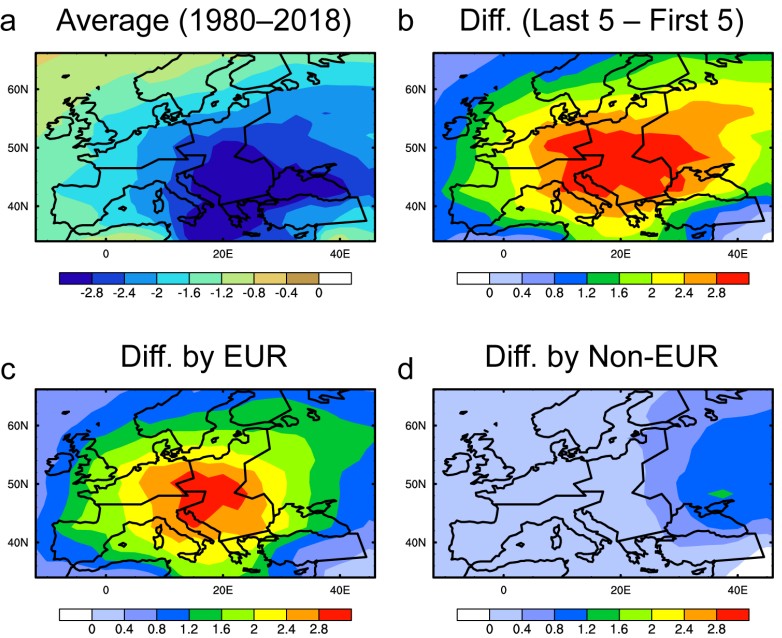

**Figure 10.** (a) Simulated annual mean DRF (W m$^{-2}$) of sulfate averaged over 1980–2018 and (b) the difference in sulfate DRF between 1980–1984 and 2014–2018. The contributions of European and non-European emissions to the difference are given in (c) and (d), respectively.





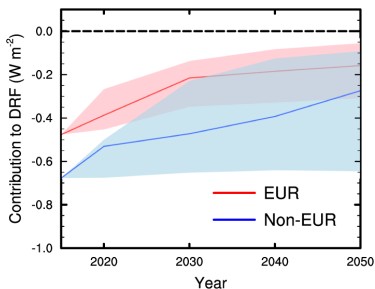

**Figure 11.** Time series (2015–2050) of estimated annual mean sulfate DRF over Europe contributed by European and non-European emissions. Lines and areas represent median values and minimum-to-maximum ranges of the estimated sulfate DRF from eight SSP scenarios, including SSP1-1.9, SSP1-2.6, SSP2-4.5, SSP3-7.0, SSP4-3.4, SSP4-6.0, SSP5-3.4, and SSP5-8.5. Future DRF of sulfate aerosol over Europe is estimated by scaling historical mean (1980–2018) sulfate DRF using the ratio of SSPs future $SO_2$ emissions to historical emissions assuming a linear response of DRF to regional emissions.