# Peer review of "Trends and source apportionment of aerosols in Europe during 1980–2018"

_Atmospheric Chemistry and Physics, 2019_

## Referee Comment (RC1) · Anonymous Referee #1 · 12 Nov 2019

The study by Yang et al., attempted to quantify the contribution of major source regions in the world towards aerosol loading in Europe. The study has certain flaws which needs to be addressed before it can be accepted for publication at ACP. Line 163: Any specific reason on why future DRF due to aerosols other than sulphate was not estimated in this study for future. If not, I would suggest doing the same. Line 175: Why was nitrate and ammonium aerosols were not considered in this study? I would suggest including nitrate at least. Line 218: I would strongly suggest to not compare the sum of BC,OC and sulphate with PM2.5 from observations until aeolian dust, sea salt, nitrate and ammonium are presented/simulated. Additionally, I feel it is meaningless to compare the model AOD (without components like nitrate, ammonium) with AERONET. Line 220: Any specific reasons on why the model does not have the capability to simu-

late ammonium and nitrate aerosols. I strongly suggest the authors to include a section on seasonal source-receptor relationship for Europe supported by meteorological factors (like wind directions). I understand it is computationally expensive to carry out this for all the years considered in the study. However, performing seasonal analysis for a single representative year would suffice.

---

## Referee Comment (RC2) · Anonymous Referee #3 · 13 Nov 2019

This study examined source apportionment of aerosols in Europe over 1980-2018 using the Community Atmosphere Model version 5 with an Explicit Aerosol Source Tagging technique (CAM5-EAST). They found that the near-surface total mass concentration of sulfate, black carbon and primary organic carbon had a 62% decrease and aerosols from foreign sources became increasingly important to air quality in Europe. They also estimated that contributions to the sulfate radiative forcing over Europe from both European local emissions and non-European emissions would decrease at a comparable rate in the next three decades. The CAM5-EAST model showed its advantage in simulating the aerosol source-receptor relationship within one model simulation. The topic is interesting and the manuscript is well organized. I suggest it published in the journal after addressing my minor comments below.

[Figure]

The authors examined sulfate, black carbon and organic carbon aerosols in this study. Why did the author exclude other aerosols like nitrate in the simulation?

There seems a lot difference between the source attribution to near-surface concentration and column loading, as demonstrated in Figure 6. Thus, it would be more clear to directly show the transport pattern and source contributions near surface as well as those at higher altitude.

In Figure 11, the areas represent minimum-to-maximum ranges. Is there a possibility that one SSP scenario produces a minimum decrease in EUR contribution and a maximum decrease in Non-EUR contribution?

What is the advantage of using CAM5-EAST rather than CAMx or CMAQ mentioned in the introduction section?

The author analyzed annual averaged source contributions in this study. How is the source-receptor relationship in different seasons? Are they the same as the annual mean results?

Page 11: What is temporal resolution of the observational data?

Fig.5: specify the abbreviations in the figure

[Figure]

---

## Author Response (AR1)

**Responses to Reviewer #1**

The study by Yang et al., attempted to quantify the contribution of major source regions in the world towards aerosol loading in Europe. The study has certain flaws which needs to be addressed before it can be accepted for publication at ACP.

We thank the reviewer for all the insightful comments. Below, please see our point-by-point response (in blue) to the specific comments and suggestions and the changes that have been made to the manuscript, in an effort to take into account all the comments raised here.

Line 163: Any specific reason on why future DRF due to aerosols other than sulphate was not estimated in this study for future. If not, I would suggest doing the same.

Response:

Here in this study we focus both historical and future sulfate DRF rather than other aerosol species. Aerosol DRF is defined in this study as the difference in clear-sky radiative fluxes at the top of the atmosphere between two diagnostic calculations in the radiative transfer scheme with and without specific aerosol species accounted, respectively. Therefore, the DRF estimate requires additional calculations of radiative fluxes. In default CAM5, less than 9 additional radiation calculations are allowed. Since that the simulation was designed to output the DRF related to emissions from Europe as well as few key source regions of the world that were used in our previous studies, it was not feasible to separate all aerosol species for the radiation diagnostic calculations. Considering that sulfate AOD accounts for the largest portion (91%) of the decrease in total combustion AOD in Europe, the sulfate DRF is calculated to roughly represent the DRF due to the total combustion AOD change.

We have now added an explanation in the revised manuscript: "Rather than sulfate, DRF of other aerosol species is not calculated in this study due to the computational limitation considering multiple source regions. However, because sulfate dominates the decrease in total combustion AOD in Europe shown below, the sulfate DRF is calculated to roughly represent the DRF caused by the total combustion AOD change."

Line 175: Why was nitrate and ammonium aerosols were not considered in this study? I would suggest including nitrate at least.

Response:

The representation of nitrate and ammonium aerosols requires many additional gas species and chemical/physical process treatments in models. Different from regional air quality models, including complex chemistry and aerosol thermodynamical equilibrium is less efficient for the long-term simulation in global
aerosol-climate models. In the 3-mode version of modal aerosol model in CAM5,
sulfate is partially neutralized by ammonium in the form of $NH_4HSO_4$, so ammonium
is effectively prescribed, but this model version cannot predict ammonium and
nitrate. In the next version of CAM6, which will be released early next year, an
advanced aerosol chemistry and microphysics module (called MOSAIC) will be
implemented to treat tropospheric trace gas photochemistry, aerosol
thermodynamics, kinetic gas-particle mass transfer and particle-phase chemistry,
particularly, for nitrate aerosol. As a next step in our research plan, we will implement
the tagging tool EAST to the new model version and analyze the source-receptor
relationship of sulfate-nitrate-ammonium in future studies.
Line 218: I would strongly suggest to not compare the sum of BC, OC and sulphate
with PM2.5 from observations until aeolian dust, sea salt, nitrate and ammonium are
presented/simulated. Additionally, I feel it is meaningless to compare the model AOD
(without components like nitrate, ammonium) with AERONET.
Response:
Thanks for the suggestion. We have removed the $PM_{2.5}$ comparison in the
manuscript and revised corresponding descriptions. For the comparison of model
and observed AOD, although the sum of sulfate (or $NH_4HSO_4$ specifically), BC, POA,
SOA, dust and sea salt cannot represent total aerosols in the real world, the
comparison is still meaningful. The purpose here is to show the decreasing trend of
AOD in Europe. Including nitrate aerosol in the simulation is unlikely to reverse the
trend. So, we decide to keep the AERONET lines in the figure.
Line 220: Any specific reasons on why the model does not have the capability to
simulate ammonium and nitrate aerosols.
Response:
Please see the responses above.
I strongly suggest the authors to include a section on seasonal source-receptor
relationship for Europe supported by meteorological factors (like wind directions). I
understand it is computationally expensive to carry out this for all the years
considered in the study. However, performing seasonal analysis for a single
representative year would suffice.
Response:
Thanks for the suggestion. We have now added the analysis of seasonal source-
receptor relationship of aerosols in Europe and the role of meteorological factors
based on an emission normalization method. Please see below:
"Source contributions to aerosols in Europe vary with season due to the
seasonality of emissions and meteorology. In general, local sources have the largest
contributions to both near-surface concentration and column burden of European
aerosols in winter and smallest contributions in summer averaged over 2010–2018
(outer rings in Figure 7). With the contributions normalized by the ratio of seasonal anthropogenic emission to annual mean for each source, the impact of emission
seasonal variation on the source contributions can be removed (inner rings in Figure
7) (Yang et al., 2019). Without the influence of emission seasonality, local source
contributions decrease in winter and increase in summer, indicating that it was the
higher local anthropogenic emissions that result in the larger local source
contributions to wintertime aerosols in Europe relative to other seasons. Sulfur
sources over oceans account for one fourth to one third of European sulfate
concentration and burden in spring likely due to the strong westerlies in this season
that transport aerosols from the North Atlantic Ocean to the Europe. Source
contributions from Russia-Belarus-Ukraine and North America to BC and POA in
Europe show strong seasonal variabilities, which can be explained by the changes in
biomass burning emissions considering its large seasonal variability."

[Figure]

**Figure 6.** Relative contributions (%) by emissions from major tagged source regions to near-surface concentrations (Conc.) and column burdens of December-January-February (DJF), March-April-May (MAM), June-July-August (JJA) and September-October-November (SON) mean sulfate, BC and POA over the Europe averaged over 2010–2018. Outer rings represent the modeled values and the relative contributions in inner rings is calculated based on absolute values normalized by the ratio of seasonal emission to annual mean. Values larger than 5% are marked.

Reference:
Yang, Y., Smith, S. J., Wang, H., Lou, S., and Rasch, P. J.: Impact of anthropogenic emission injection height uncertainty on global sulfur dioxide and aerosol distribution, J. Geophys. Res.-Atmos., 124, 4812–4826. https://doi.org/10.1029/2018JD030001, 2019.

**Responses to Reviewer #3**

This study examined source apportionment of aerosols in Europe over 1980-2018 using the Community Atmosphere Model version 5 with an Explicit Aerosol Source Tagging technique (CAM5-EAST). They found that the near-surface total mass concentration of sulfate, black carbon and primary organic carbon had a 62% decrease and aerosols from foreign sources became increasingly important to air quality in Europe. They also estimated that contributions to the sulfate radiative forcing over Europe from both European local emissions and non-European emissions would decrease at a com- parable rate in the next three decades. The CAM5-EAST model showed its advantage in simulating the aerosol source-receptor relationship within one model simulation. The topic is interesting and the manuscript is well organized. I suggest it published in the journal after addressing my minor comments below.

We thank the reviewer for all the insightful comments. Below, please see our point-by-point response (in blue) to the specific comments and suggestions and the changes that have been made to the manuscript, in an effort to take into account all the comments raised here.

The authors examined sulfate, black carbon and organic carbon aerosols in this study. Why did the author exclude other aerosols like nitrate in the simulation?
Response:
The representation of nitrate and ammonium aerosols requires many additional gas species and chemical/physical process treatments in models. Different from regional air quality models, including complex chemistry and aerosol thermodynamical equilibrium is less efficient for the long-term simulation in global aerosol-climate models. In the 3-mode version of modal aerosol model in CAM5, sulfate is partially neutralized by ammonium in the form of $NH_4HSO_4$, so ammonium is effectively prescribed, but this model version cannot predict ammonium and nitrate. In the next version of CAM6, which will be released early next year, an advanced aerosol chemistry and microphysics module (called MOSAIC) will be implemented to treat tropospheric trace gas photochemistry, aerosol thermodynamics, kinetic gas-particle mass transfer and particle-phase chemistry, particularly, for nitrate aerosol. As a next step in our research plan, we will implement the tagging tool EAST to the new model version and analyze the source-receptor relationship of sulfate-nitrate-ammonium in future studies.

There seems a lot difference between the source attribution to near-surface concentration and column loading, as demonstrated in Figure 6. Thus, it would be more clear to directly show the transport pattern and source contributions near surface as well as those at higher altitude.
Response:

Thanks for the suggestion. We have now added the horizontal distribution of sulfate-BC-POA concentrations at the surface and 500 hPa, originating from the major tagged source regions, as shown below.

"The transboundary and intercontinental transport of aerosols occur most frequently in the free troposphere rather than near the surface, as horizonal transport pathways at the surface and 500 hPa are indicated on the spatial distribution map of the relative contributions shown in Figures S2 and S3. This also leads to larger relative contributions from non-European sources to aerosol column burdens than to the near-surface concentrations."

[Figure]

**Figure S2.** Relative contributions (%) to annual mean near-surface concentrations of sulfate-BC-POA from the major tagged source regions including Europe (EUR), North America (NAM), North Africa (NAF), the Middle East (MDE), East Asia (EAS), Russia-Belarus-Ukraine (RBU), Non-Arctic/Antarctic Ocean (OCN) and other (OTH) regions averaged over 2010–2018.

[Figure]

**Figure S3.** Relative contributions (%) to annual mean concentrations of sulfate-BC-POA at 500 hPa from the major tagged source regions including Europe (EUR),

North America (NAM), North Africa (NAF), the Middle East (MDE), East Asia (EAS),
Russia-Belarus-Ukraine (RBU), Non-Arctic/Antarctic Ocean (OCN) and other (OTH)
regions averaged over 2010–2018.
In Figure 11, the areas represent minimum-to-maximum ranges. Is there a possibility
that one SSP scenario produces a minimum decrease in EUR contribution and a
maximum decrease in Non-EUR contribution?
Response:
We have now plotted the figure for each SSP scenario individually in Figure S4.
All SSPs show that non-European contributions change in a magnitude similar to
that of European local emissions.

[Figure]

**Figure S4.** Time series (2015–2050) of estimated annual mean sulfate DRF over
Europe contributed by European and non-European emissions from eight SSP
scenarios, including SSP1-1.9, SSP1-2.6, SSP2-4.5, SSP3-7.0, SSP4-3.4, SSP4-
6.0, SSP5-3.4, and SSP5-8.5. Future DRF of sulfate aerosol over Europe is
estimated by scaling historical mean (1980–2018) sulfate DRF using the ratio of
SSPs future $SO_2$ emissions to historical emissions assuming a linear response of
DRF to regional emissions.

What is the advantage of using CAM5-EAST rather than CAMx or CMAQ mentioned
in the introduction section?
Response:
Influences of remote sources simulated in regional air quality models such as
CAMx and CAMQ largely depend on the boundary conditions. They cannot tag and
track the emissions outside the regional domain. As we discuss in the text,
"However, due to the limitation in domain size of regional air quality models,
contributions of intercontinental transport from sources outside the domain are
difficult to be accounted." CAM5-EAST is a global model with aerosol tagging that
has been used to examine the transboundary and transcontinental transport of
aerosols in previous studies (Yang et al., 2018a,b).
The author analyzed annual averaged source contributions in this study. How is the
source-receptor relationship in different seasons? Are they the same as the annual
mean results?
Response:
Thanks for the suggestion. We have now added the analysis of seasonal source-
receptor relationship of aerosols in Europe and the role of meteorological factors
based on an emission normalization method. Please see below:
"Source contributions to aerosols in Europe vary with season due to the
seasonality of emissions and meteorology. In general, local sources have the largest
contributions to both near-surface concentration and column burden of European
aerosols in winter and smallest contributions in summer averaged over 2010–2018
(outer rings in Figure 7). With the contributions normalized by the ratio of seasonal
anthropogenic emission to annual mean for each source, the impact of emission
seasonal variation on the source contributions can be removed (inner rings in Figure
7) (Yang et al., 2019). Without the influence of emission seasonality, local source
contributions decrease in winter and increase in summer, indicating that it was the
higher local anthropogenic emissions that result in the larger local source
contributions to wintertime aerosols in Europe relative to other seasons. Sulfur
sources over oceans account for one fourth to one third of European sulfate
concentration and burden in spring likely due to the strong westerlies in this season
that transport aerosols from the North Atlantic Ocean to the Europe. Source
contributions from Russia-Belarus-Ukraine and North America to BC and POA in
Europe show strong seasonal variabilities, which can be explained by the changes in
biomass burning emissions considering its large seasonal variability."

[Figure]

DJF  MAM  JJA  SON

Sulfate Conc.

Sulfate Burden

BC Conc.

BC Burden

POA Conc.

POA Burden

■ EUR ■ NAM ■ NAF ■ MDE ■ EAS ■ RBU ■ OTH ■ OCN

**Figure 6.** Relative contributions (%) by emissions from major tagged source regions to near-surface concentrations (Conc.) and column burdens of December-January-February (DJF), March-April-May (MAM), June-July-August (JJA) and September-October-November (SON) mean sulfate, BC and POA over the Europe averaged over 2010–2018. Outer rings represent the modeled values and the relative contributions in inner rings is calculated based on absolute values normalized by the ratio of seasonal emission to annual mean. Values larger than 5% are marked.

Page 11: What is temporal resolution of the observational data?
Response:
    We have now added a description that "EMEP (European Monitoring and Evaluation Programme, http://www.emep.int) networks provide daily near-surface aerosol concentrations in Europe. The annual mean of daily observations is used to evaluate the model performance in this study."

Fig.5: specify the abbreviations in the figure
Response:
Revised.
Reference:

[revised manuscript text omitted]